# Efficient Mask Optimization for DMD-Based Maskless Lithography Using a Genetic–Hippo Hybrid Algorithm

**DOI:** 10.3390/mi16121333

**Published:** 2025-11-27

**Authors:** Zhiyong Chen, Chi Tu, Haifeng Sun, Xia Kang, Junbo Liu, Song Hu

**Affiliations:** 1National Key Laboratory of Optical Field Manipulation Science and Technology, Chinese Academy of Sciences, Chengdu 610209, China; chenzhiyong23@mails.ucas.ac.cn (Z.C.); tuchi23@mails.ucas.ac.cn (C.T.); sunhaifeng@ioe.ac.cn (H.S.); kangxia@ioe.ac.cn (X.K.); husong@ioe.ac.cn (S.H.); 2State Key Lab of Optical Technologies on Nano-Fabrication and Micro-Engineering, Chinese Academy of Sciences, Chengdu 610209, China; 3Institute of Optics and Electronics, Chinese Academy of Sciences, Chengdu 610209, China; 4University of Chinese Academy of Sciences, Beijing 100049, China

**Keywords:** maskless lithography, Genetic–Hippo hybrid optimization algorithm, optical proximity effect correction

## Abstract

Mask optimization is a critical technique for enhancing imaging performance in digital micromirror device (DMD)-based maskless lithography. Conventional algorithms, however, often suffer from slow convergence and limited adaptability, particularly when handling complex multi-feature mask patterns. To address these challenges, this study proposes a hybrid Genetic–Hippo Optimization (GA-HO) algorithm that integrates the global exploration capability of the Genetic Algorithm (GA) with the local exploitation efficiency of the Hippocampus Optimization (HO) Algorithm. The approach employs grayscale modulation for adaptive mask optimization and introduces a global–local cyclic search mechanism to balance exploration and exploitation throughout the optimization process. Simulation results demonstrate that the GA-HO hybrid algorithm achieves a more pronounced improvement in overall optimization performance compared with the standard GA. In complex multi-line mask optimization, the standard GA achieves approximately a 18% enhancement in optimization accuracy, whereas the GA-HO algorithm achieves around a 30% improvement. Moreover, the GA-HO algorithm exhibits a smoother convergence curve, greater stability, and superior robustness. The hybrid method effectively suppresses linewidth variations and corner distortions caused by optical proximity effects (OPE), maintaining high imaging fidelity and stable optimization outcomes even under challenging mask conditions. Overall, the proposed GA-HO framework demonstrates excellent efficiency, adaptability, and precision, providing a reliable and high-performance solution for DMD-based maskless lithography. This work offers a strong theoretical and algorithmic foundation for advancing high-resolution, high-efficiency, and low-cost micro/nanofabrication technologies, highlighting the potential of heuristic hybrid optimization strategies for practical lithography applications.

## 1. Introduction

With the continuous advancement of micro/nanofabrication technologies, high-precision micro–nanostructures-represented by integrated circuits [1,2], micro-optical components [3,4], bioinspired structures [5,6], and metasurfaces [7,8] have found extensive applications in aerospace [9], defense [10], and advanced optoelectronic devices [11,12]. However, conventional mask-based photolithography faces numerous challenges in achieving high-resolution patterning. Its dependence on physical masks leads to long fabrication cycles and high costs, as well as issues such as pattern degradation and alignment errors, which significantly limit its applicability in rapid prototyping and customized manufacturing.

To overcome these limitations, maskless lithography has emerged as a promising alternative. As illustrated in Figure 1, the distinction between conventional lithographic imaging and DMD-based maskless lithography is highlighted. In digital micromirror device (DMD)-based maskless lithography systems, programmable micromirrors act as dynamic virtual masks to directly generate exposure patterns, eliminating the need for physical photomasks. This approach offers high process flexibility and substantially reduced mask fabrication costs. It also enables real-time adjustment of exposure patterns and rapid verification of design layouts, making it highly suitable for rapid prototyping and personalized manufacturing [13]. Nevertheless, the intrinsic pixel size of DMDs imposes constraints on both resolution and exposure speed. Moreover, the discrete nature of DMD micromirrors introduces pixelated edges, resulting in edge roughness in fine pattern fabrication. When the feature size approaches the system’s resolution limit, optical proximity effects (OPEs) become the dominant bottleneck for imaging performance [14].

To address this challenge, optical proximity correction (OPC) has become a key technique for improving imaging accuracy in DMD-based maskless lithography. Early OPC studies primarily relied on empirical models to modify mask patterns [15]. Later, researchers developed DMD-specific OPC approaches [16], which adjusted the grayscale distribution of digital masks through simulation, successfully reducing edge displacement and corner rounding between the designed layout and the resist pattern. More recently, intelligent optimization algorithms have been introduced into DMD-based OPC. Yang et al. [17] proposed a GA-based OPC method that adaptively optimized pixel-level grayscale, significantly improving pattern fidelity. Zhang et al. [18] developed an improved particle swarm optimization algorithm that enhanced both search efficiency and imaging quality. Meanwhile, deep learning-based approaches have also emerged [19]; for instance, U-net architectures achieved pixel-level OPE correction, markedly reducing layout-to-pattern errors. Chan et al. [20] introduced a deep-learning-driven digital inverse lithography technique (DDILT), which optimized DMD modulation coefficients for OPE correction, greatly improving pattern printability. These studies—combining physical modeling with algorithmic optimization—have provided valuable insights into enhancing pattern fidelity in DMD lithography.

Despite these advances, traditional OPC strategies still face limitations in DMD lithography. First, conventional algorithms typically require extensive iterative searches, resulting in high computational cost and slow convergence, with a strong tendency to become trapped in local optima. Second, the discrete and finite grayscale characteristics of DMD pixels complicate the optimization process, as continuous grayscale models used in conventional lithography are not directly applicable. Additionally, non-uniform illumination and complex optical transfer functions in scanning exposure systems make rule-based correction methods difficult to generalize across different pattern types. Consequently, existing OPC approaches struggle to balance optimization efficiency and imaging accuracy, highlighting the need for new algorithms that can expand the search space and improve overall optimization performance.

To address these issues, heuristic metaheuristic algorithms—which do not require explicit global problem information and are suitable for large-scale nonlinear optimization—have attracted increasing attention. These algorithms can efficiently explore complex search spaces through population-based cooperative search, effectively balancing global exploration and local exploitation to approximate global optima. The recently developed Hippo Optimization (HO) algorithm [21] provides a promising foundation for this task. Inspired by the collective behavior of hippos, the HO algorithm employs a three-phase update strategy and has demonstrated excellent performance across various benchmark functions. It combines strong global exploration with robust local convergence, making it well-suited for complex grayscale mask optimization.

Building upon these insights, this paper proposes a Genetic–Hippo Hybrid Optimization (GA-HO) algorithm for adaptive grayscale mask optimization in DMD-based maskless lithography. The proposed method leverages the global search capability of the Genetic Algorithm (GA) and the local exploitation strength of the Hippo Optimization Algorithm (HO), significantly improving optimization efficiency while maintaining imaging fidelity. The main contributions of this study are summarized as follows:

(1) Innovative Algorithm Framework Design: The core contribution of this study lies in the development of a hybrid optimization framework that integrates the Genetic Algorithm (GA) and the Hippo Optimization (HO) algorithm. This framework effectively combines the strong global stochastic exploration capability of GA with the efficient local exploitation ability of HO. Through the coordinated cooperation of global and local searches, the proposed method achieves global–local collaborative optimization of the grayscale distribution in DMD masks, effectively overcoming the limitations of conventional single algorithms that often suffer from slow convergence or entrapment in local optima when dealing with complex optimization problems.

(2) Efficient Performance Optimization Mechanisms: To further enhance the stability and convergence efficiency of the algorithm, this study introduces an adaptive elite preservation mechanism and a multi-leader guidance mechanism based on the HO algorithm. The adaptive elite preservation mechanism ensures that high-quality solutions are retained throughout the evolutionary process, maintaining the overall quality of the population. Meanwhile, the multi-leader guidance mechanism—modeled after the social behavior of hippo groups—utilizes elite individuals to direct the population’s evolutionary trajectory. These mechanisms jointly improve algorithmic stability and convergence, significantly strengthening the local search capability. As a result, the algorithm can approach the global optimum more rapidly and stably within complex solution spaces.

(3) Comprehensive Simulation Validation: Multiple comparative simulation experiments were conducted to validate the effectiveness of the proposed algorithm. The results demonstrate that the GA-HO hybrid algorithm maintains high optimization accuracy and efficiency even under complex multi-line mask scenarios. Quantitative comparisons show that, compared with the standard GA, the proposed hybrid algorithm achieves significant improvements in both optimization precision and convergence stability. These findings confirm the effectiveness and superiority of the hybrid optimization strategy, providing a stable, efficient, and intelligent optimization solution for high-resolution DMD-based maskless lithography, with strong theoretical significance and practical application value. A basic sketch of DMD-based maskless lithography is shown in Figure 2.

## 2. Theory and Modeling

### 2.1. Spatial Imaging Model for DMD-Based Lithography Based on Abbe Theory

In a DMD-based lithography system, when the distance between the DMD and the exposure plane, the micromirror size, and the incident wavelength satisfy the Fraunhofer diffraction condition, the diffraction characteristics can be approximately described using Fourier optics [22]. Assuming the incident light is a monochromatic plane wave, the complex amplitude distribution on the DMD surface can be expressed as:
(1)$$U_{DMD}\left(x,y\right)=A\left(x,y\right)\cdot e^{j\varphi \left(x,y\right)}$$ where
A(x,y) denotes the reflection amplitude and
 ϕ(x,y) is the phase distribution. For an array of micromirrors in the ON state, the amplitude is controlled by grayscale values and can be written as:
(2)$$A\left(x,y\right)=\sum_{m,n}G_{m,n}\cdot rect(\frac{x-md}{d})\cdot rect(\frac{y-md}{d})$$ where
$\;G_{m,n}$ is the grayscale weight corresponding to the
 (m,n)-th micromirror, and
d  is the center-to-center pitch of a single micromirror. These grayscale weights are realized through pulse-width modulation and time-division modulation, enabling 16-level grayscale control on the DMD.

After focusing by the optical system, the complex amplitude on the image plane is obtained by convolution of the DMD diffraction field with the system point spread function (PSF):
(3)$$U_{img}(x,y)=U_{DMD}(x,y)*h(x,y)$$ where
h(x,y) represents the system’s impulse response. The final exposure intensity is then given by:
(4)$$I(x,y)=|U_{img}(x,y){|}^{2}$$

This model establishes a quantitative relationship between the DMD pixel grayscale and the resulting exposure distribution, providing a solid foundation for evaluating the fitness of mask patterns in subsequent grayscale optimization algorithms.

### 2.2. Optical Field Amplitude Modulation Model Based on DMD

In digital lithography, when the distance
z between the DMD and the photoresist, the micromirror width
w, and the incident wavelength
λ satisfy
$z\geq \pi w^{2}/\lambda$, the diffraction can be analyzed using Fraunhofer diffraction theory. Accordingly, when a monochromatic plane wave illuminates the ON-state micromirror array at a spatial angle
θ, the complex amplitude on the DMD window surface can be expressed as:
(5)$$f\left(x,y\right)=\left\{\left[rect\left(\frac{x}{b}\right)rect\left(\frac{y}{b}\right)\right]\exp\left[i2\pi k\left(x+y\right)\right]\right\}\times \sum_{m=-\frac{M}{2}}^{\frac{M}{2}-1}\sum_{n=-\frac{N}{2}}^{\frac{N}{2}-1}\sigma \left(x+md,y+nd\right)$$

### 2.3. Photoresist Exposure Model

To facilitate algorithm implementation and simulation, the photoresist exposure-development process in this work is simplified. Traditional models typically employ a continuous nonlinear dose-development rate relationship to describe the variation in resist etch depth with exposure dose. However, these models involve numerous parameters and high computational complexity, which hinder the rapid convergence of optimization algorithms in large-scale mask optimization.

Therefore, a threshold-based simplified model [23] is adopted in this study. The normalized exposure intensity is used as an approximate measure of the dose, and the development state of the photoresist is determined by a fixed threshold. The model can be expressed as follows:
(6)$$\left(x,y\right)=\left\{\begin{array}{c} 1,I(x,y)\geq T\\ 0,I\left(x,y\right)<T \end{array}\right.$$ where
I(x,y) denotes the normalized exposure intensity distribution, and
T represents the photoresist exposure threshold (set to 0.7 in this study). Regions where the local exposure intensity exceeds the threshold are considered fully developed, while areas below the threshold remain undeveloped. This model significantly simplifies the computation while preserving interpretability of the developed features, making it suitable for rapid evaluation of pattern fidelity and for fitness calculations in optimization algorithms.

During simulation, the exposure intensity
I(x,y) is obtained from the Abbe imaging model, normalized, and then input into the threshold function to generate a binary development pattern. The resulting photoresist profile is subsequently compared with the target mask to calculate pattern-matching fidelity, providing a quantitative metric for the Genetic–Hippo Hybrid Optimization (GA-HO) algorithm.

## 3. Maskless Optimization Based on Genetic–Hippo Hybrid Algorithm

### 3.1. Genetic Algorithm

The genetic algorithm (GA) is a stochastic optimization method inspired by biological evolution, with its core principle derived from the concept of “survival of the fittest” [24]. GA simulates the reproduction and evolution of a population to gradually approach the global optimum in a complex solution space [25]. In DMD mask optimization, each individual represents a potential grayscale mask matrix
$M_{i}(x,y)$. The fitness of each individual is evaluated based on the degree of match between its simulated aerial image and the target pattern
T(x,y). The basic GA procedure includes encoding, fitness evaluation, selection, crossover, and mutation. First, Encoding and Initial Population Generation: Mask grayscale values are quantized into 16 levels, with each pixel corresponding to a gene. Several candidate solutions are generated by random perturbation on the initial mask to form the initial population, ensuring diversity in the search. Second, Fitness Function Definition: the objective of the fitness function is to minimize the error between the lithographic imaging pattern and the target mask pattern. To quantitatively evaluate the optimization performance, image subtraction is used to calculate the degree of similarity between the exposure pattern and the mask pattern. The matching rate
$F_{i}$ can be defined as [17]:
(7)$$F_{i}=1-\frac{\sum_{x,y}abs\left(T\left(x,y\right)-I_{0}\left(x,y\right)\right)}{\sum_{x,y}I_{0}\left(x,y\right)}\times 100\%$$ where
T(x,y) is the target mask pattern and
$I_{0}(x,y)$ is the exposure pattern. A larger
$F_{i}$ indicates closer correspondence between the mask and the target image. Last, Genetic Operations: Tournament selection is used, where the fittest individual is chosen from a set of randomly selected candidates. Then Crossover: Arithmetic crossover is applied:
(8)$$C=\alpha P_{1}+(1-\alpha )P_{2}$$ where
$P_{1}$ and
$P_{2}$ are parent individuals, and
α∈[0,1] is a random mixing factor.

Mutation: Adaptive Gaussian perturbation is applied:
(9)$$M^{\prime }=M+\sigma_{t}\cdot N(0,1)$$ where
$\sigma_{t}$ decays gradually with iterations to balance exploration and convergence.

Although GA excels at global search, it suffers from slow convergence and susceptibility to local optima. Therefore, this study introduces the Hippo Optimization (HO) algorithm as a local refinement module to enhance overall performance.

### 3.2. Hippo Optimization Algorithm

HO is a novel swarm intelligence optimization method inspired by the social behavior of hippos [21]. The algorithm mathematically models ecological mechanisms such as group collaboration, predator defense, and escape behavior, achieving a dynamic balance between global and local search.

Multi-Leader Guidance Mechanism.

In each generation, several leaders are selected based on fitness, with selection probability defined as:
(10)$$P_{i}=\frac{f_{i}^{\beta }}{\sum_{k=1}^{E}f_{k}^{\beta }}$$ where
$f_{i}$ is the fitness of the
i-th elite,
E is the number of elites, and
β is the selection pressure coefficient. This ensures that high-quality individuals are more likely to guide the search.

2.Directed Gene Replacement:

The HO module randomly selects gene segments from leaders to replace parts of ordinary individuals:
(11)$$X_{i}^{\left(t+1\right)}=X_{i}^{\left(t\right)}+\eta \cdot (L_{j}^{\left(t\right)}-X_{i}^{\left(t\right)})$$ where
$X_{i}^{\left(t\right)}$ is the
i-th individual,
$L_{j}^{\left(t\right)}$ is a randomly chosen leader, and
η is the learning rate. This process acts as a directed mutation, allowing the population to converge toward superior solutions without losing diversity.

3.Local Escape Mechanism.

When stagnation is detected, a Cauchy-distributed perturbation is applied near the current solution:
(12)$$X_{i}^{\left(t+1\right)}=X_{i}^{\left(t\right)}+\delta \cdot \mathrm{Cauchy}(0,1)$$ where
δ is the perturbation amplitude. This helps the algorithm escape local optima.

Through these three mechanisms, the HO module provides local refinement in the hybrid algorithm, complementing GA’s global exploration.

### 3.3. GA-HO Hybrid Optimization Structure

The GA-HO hybrid algorithm integrates GA’s global search with HO’s local exploitation, achieving efficient mask optimization through multi-level interaction [17,21]. The algorithm structure is illustrated in Figure 3.

The algorithm primarily consists of the following stages:Initialization: Randomly generate several mask grayscale matrices as the initial population and calculate their imaging fitness.Genetic Evolution: Perform selection, crossover, and mutation operations to produce a new generation of individuals.Hippo Enhancement: Select leaders from elite individuals and perform gene-guided local search.Adaptive Perturbation: If no improvement in the best fitness is observed over consecutive generations, trigger the drought (perturbation) mechanism to restore population diversity.Convergence Check: Terminate the algorithm when the rate of change in the best fitness falls below a preset threshold or the maximum number of iterations is reached.

### 3.4. Convergence and Adaptive Control Mechanisms

To balance optimization efficiency and search stability, three adaptive control strategies are implemented [21,26]:

First, adaptive Elite Retention: The proportion of elites is dynamically adjusted based on population diversity:
(13)$$p_{E}=p_{E,min}+(p_{E,max}-p_{E,min})\cdot \frac{D_{t}}{D_{0}}$$ where
$D_{t}$ is the current population diversity and
$D_{0}$ is the initial diversity.

Second, Evolution Stagnation Detection: A sliding window monitors the rate of change of the best fitness:
(14)$$S_{t}=\frac{1}{W}\sum_{i=t-W}^{t}|F_{i}-F_{i-1}|$$ when
$S_{t}<\delta_{s}$, the local perturbation mechanism is triggered.

Last, Cauchy Perturbation Strategy: Cauchy-distributed perturbations replace Gaussian noise to enhance the ability to escape local optima. These mechanisms allow the algorithm to adaptively adjust the balance between exploration and exploitation, ensuring stable convergence in complex mask optimization problems [27].

### 3.5. Mathematical Analysis and Synergistic Mechanism of the GA-HO Hybrid Optimization Algo-Rithm

The core advantage of the GA-HO hybrid algorithm lies in its cooperative “global exploration + local refinement” mechanism, which makes it particularly suitable for the high-dimensional and highly non-convex nature of DMD maskless lithography. In DMD mask design, each micromirror corresponds to a discrete grayscale value, forming a multi-dimensional discrete search space. Due to the optical proximity effect (OPE), this space contains many local optima, making it difficult for a single optimization method to achieve both global exploration and fine local adjustment.

The Genetic Algorithm (GA) performs global search through selection, crossover, and mutation. It can explore a wide solution space, but its search process is largely random, which often results in slow convergence. Moreover, GA does not include a strong local refinement mechanism, so its ability to correct complex imaging errors such as OPE is limited. In contrast, local optimization algorithms converge faster but are easily trapped in local optima. Even with disturbance strategies, their ability to escape local minima remains limited.

Although GA has been proven effective in DMD mask optimization, it still tends to suffer from loss of diversity and premature convergence during evolution [17,18]. The Hippo Optimization (HO) algorithm, inspired by cooperative, leadership, and disturbance behaviors within hippo groups, provides stronger local exploitation and directional search capability [21]. Previous studies show that HO often outperforms basic algorithms like PSO and GA, especially in convergence speed and final solution quality.

Based on these characteristics, this work integrates GA and HO to construct a global-local cooperative optimization framework: First, the GA performs global exploration and maintains search breadth. Second, the HO algorithm conducts fine-grained local refinement in promising regions. Finally, the HO algorithm further guides the population toward high-fitness regions by injecting effective gene segments from elite individuals into ordinary individuals, improving both convergence speed and final imaging quality.

Simulation results in Section 4 show that the GA-HO algorithm achieves higher optimization efficiency and better final performance than the GA alone. In contrast, the GA by itself often shows large fluctuations and easily becomes trapped in local optima. With elite preservation and adaptive diversity control, GA-HO can follow a stable “global-local cooperative” convergence path, resulting in more consistent and robust outcomes.

From a mathematical perspective, HO improves optimization mainly through its update strategy. In each generation, HO first ranks all individuals by fitness, then selects a certain proportion of elite individuals as leaders so that high-quality solutions can effectively guide the population. It then performs a directed gene-replacement step, injecting key gene segments from leaders into ordinary individuals with amplitudes controlled by a learning rate. This enables the population to move toward better solutions while maintaining diversity. When the algorithm stagnates, a Cauchy-based escape mechanism perturbs the current best solution, helping the algorithm jump out of local minima. With adaptive elite preservation and stagnation detection, GA-HO dynamically balances exploration and exploitation throughout the iteration process.

The core update formula of the algorithm can be summarized as:
(15)$$X^{\left(t+1\right)}=\left\{\begin{array}{c} \mathrm{GA\_Update}\left(X^{\left(t\right)}\right),\;if\;\Delta F_{t}>\epsilon \\ \mathrm{HO\_Enhance}(X^{\left(t\right)}),otherwise \end{array}\right.$$

In summary, the GA-HO hybrid algorithm offers three major strengths: First, Global-local cooperation: GA provides global direction, while HO performs efficient local refinement and guides the overall evolutionary trend. Second, Faster and better convergence: HO’s directional update and Cauchy perturbation accelerate convergence and improve the final solution. Finally, Improved repeatability: Adaptive elite retention ensures stable and consistent results across multiple runs.

In lithography optimization, several hybrid intelligent algorithms—such as GA-PSO, GA-DE, GA-SA, and GA-TS—have been reported [28,29,30]. Although these methods improve pattern fidelity and convergence, they still have limitations. For example: GA-PSO combines GA’s global search with PSO’s fast convergence, but it has many parameters, a more complex structure, and PSO is weaker than HO in fine local optimization. Thus, in high-dimensional discrete problems it still tends to converge prematurely and generally performs worse than GA-HO. GA-DE integrates GA’s diversity with DE’s differential mutation, improving source-mask optimization to some extent. However, both GA and DE are essentially global search strategies and lack strong local exploitation. For OPE-related problems that rely heavily on local refinement, their effectiveness is limited.

In comparison, GA-HO incorporates multi-stage local search and adaptive perturbation from HO, achieving higher matching accuracy, smoother convergence behavior, and stronger stability in high-dimensional discrete lithography optimization. Independent experiments confirm that GA-HO consistently improves both optimization speed and final image quality.

Overall, by combining GA’s global exploration with HO’s local refinement, the GA-HO framework forms a complementary optimization strategy. In complex DMD mask optimization tasks, it shows clear advantages in imaging fidelity, convergence efficiency, and stability. The algorithm can obtain high-fidelity mask solutions within limited iterations and effectively suppress OPE-induced distortions such as linewidth variation and corner rounding. Simulation results further verify that GA-HO maintains strong optimization performance even for complex patterns and high-resolution settings. Therefore, GA-HO provides a reliable and efficient optimization approach for achieving high-resolution and high-efficiency digital lithography.

## 4. Simulation Results and Analysis

### 4.1. Simulation Setup

Simulations were conducted using a DMD-based maskless lithography system modeled with the Abbe imaging theory. The system parameters are as follows: the illumination wavelength was 365 nm, the numerical aperture (NA) was 0.3, the partial coherence factor of the light source was 0.9, the photoresist exposure threshold was 0.5, and the mask image size was 16 × 16 pixels.

The key parameters of the optimization algorithm were set as follows: the population size was 25, the maximum number of iterations was 1000, the elite retention ratio was 0.1, and the proportion of HO leaders was 0.5. The simulation parameters can be found in Table 1. To ensure sufficient depth and rigor, each mask pattern was optimized using five different algorithms, and each algorithm was run ten times. The optimization performance and computation time of every run were recorded to enable a comprehensive and reliable comparative analysis.

### 4.2. Simulation Results

In this study, four representative original masks—A, B, C, and D—were selected for optimization, as shown in Figure 4. We then applied five algorithms—GA, GA-HO, GA-TS, GA-PSO, and GA-ED—to optimize these four masks and compared their results. This comparison is used to demonstrate the performance and efficiency advantages of the proposed GA-HO algorithm.

As shown in Figure 4a, Mask A is a fairly complex pattern made up of many closely spaced parallel lines with different lengths. The line widths and gaps are very close to the resolution limit of the lithography system, so the pattern is easily affected by optical proximity effects during imaging and tends to become distorted. The contour comparison in Figure 5 and the exposure comparison in Figure 6 also show clear edge shifts and rounded corners between the aerial image and the target mask before optimization. Its initial matching rate is only 52.70%, which means the pattern fidelity is bad.

The change in the MR curve during the iteration process is shown below. From the Figure 7, we can see that the GA converges at around 300 generations. The average matching rate over the ten runs is 61.97%, with the highest reaching 63.86%. Since the program uses GPU acceleration, the overall optimization process runs faster, with an average runtime of 55.31 s.

Next, the GA-TS algorithm was applied to optimize the mask. Figure 8 shows the contour comparison and the exposure pattern obtained using this method. As seen Figure 9 GA-TS converges at around 200 generations. Because the tabu list in TS helps the algorithm avoid repeated and ineffective searches, the overall convergence speed is faster than that of the standard GA. The average runtime is 48.82 s, and the mean matching rate across the ten optimization runs is 61.72%.

It is worth noting that GA-TS does not fundamentally enhance global or local search capability. The role of TS here is mainly to impose constraints on the GA process, preventing unnecessary iterations and accelerating convergence. Therefore, this hybrid approach essentially improves optimization efficiency, while its contribution to the final optimization quality is relatively limited.

Next, the GA-DE algorithm was applied to optimize the mask. Figure 10 shows the contour comparison and exposure patterns obtained using this method. As seen in Figure 11, the convergence trend of GA- DE is very similar to that of GA, and it also reaches convergence at around 400 generations.

**Figure 10 micromachines-16-01333-f010:**
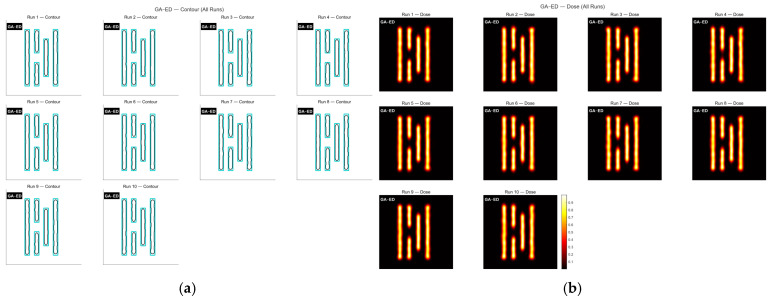
The contour comparison (**a**) and exposure patterns (**b**) obtained from ten optimization runs of the GA-DE algorithm on Mask A.

**Figure 11 micromachines-16-01333-f011:**
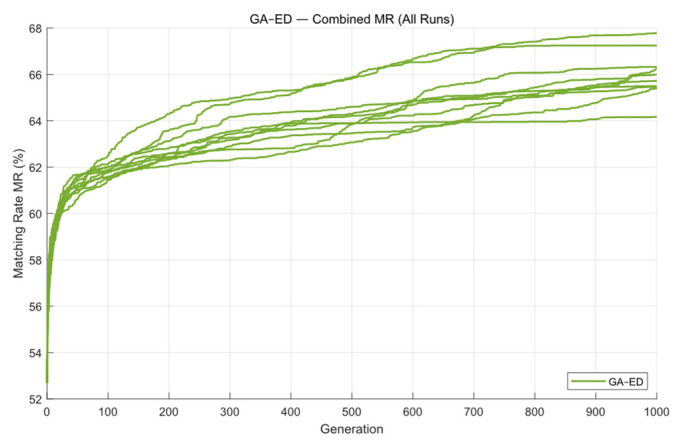
The global MR curves obtained from ten optimization runs of the GA-DE algorithm on Mask A.

It is worth noting that DE (Differential Evolution) is also a global optimization algorithm. Like GA, it uses a population-based search, but updates individuals through “differential mutation and selection,” which often gives it better numerical stability and stronger performance in continuous parameter spaces. However, this strength overlaps with GA’s global search ability rather than complementing it. In other words, both algorithms focus mainly on global exploration, and neither provides a clear improvement in local refinement.

Because of this overlap, GA- DE does not form the kind of “global + local” cooperative effect seen in GA-HO, so the final optimization quality is not substantially better than using the GA alone. In addition, the extra differential operations and trial vectors introduced by ED increase the overall computation cost.

Overall, the GA- DE algorithm has an average runtime of about 61.98 s, and the mean matching rate over ten runs is 66.02%.

Next, the GA-PSO algorithm was applied to optimize the mask. Following the method described in [28], the first 300 generations used GA for global search, and the remaining 700 generations switched to PSO for local optimization. Figure 12 shows the contour comparison and exposure patterns produced by this algorithm. The convergence process is given in Figure 13. As shown in the figure, the first 200 generations behave almost the same as the standard GA, and the algorithm reaches an initial stable state around this point, producing results similar to GA. After the 300th generation, PSO begins to take over for local refinement.

From the convergence curves, it can be seen that PSO provides a short period of local improvement between roughly 200 and 400 generations. However, this refinement is not always effective. The generation at which PSO begins to improve the result varies from run to run, and nearly half of the PSO iterations bring no actual improvement at all. This indicates that the GA-PSO algorithm is not very stable—its performance is sensitive to the initial population and randomness, and it tends to become stuck in local optima, resulting in weaker overall robustness.

Based on the experimental results, the GA-PSO algorithm has an average runtime of about 53.21 s, and the mean matching rate over ten runs reaches 68.13%. Both the optimization efficiency and the final imaging quality show a clear improvement compared with the other algorithms. However, the stability of GA-PSO is quite poor, and the variance of its optimization results is very large.

Finally, the proposed GA-HO hybrid algorithm was used to optimize the mask. Figure 14 shows the contour comparison and exposure patterns produced by this method, and Figure 15 presents the convergence process. From the figure, we can see that the “leader” step in HO guides the population toward better solutions in each generation, allowing GA-HO to reach a good level of convergence very early. Compared with the previous algorithms, GA-HO keeps a higher and steadier improvement rate from the start. With the added self-learning and small disturbance steps, the algorithm does not become stuck or waste many iterations, so the whole convergence curve becomes smoother and more reliable.

It is also clear that GA-HO can continue improving the matching rate in the later iterations. The earlier algorithms usually show little improvement once the population becomes too similar, but GA-HO can still make useful local adjustments because HO has strong local refinement ability. This allows the mask to keep improving even in the late stage. Overall, GA-HO combines the global search of GA with the fine-tuning ability of HO, giving higher efficiency, more stable convergence, and better final imaging quality.

The average runtime of GA-HO is about 50.13 s, and the mean matching rate over ten runs reaches 82.39%. The optimization efficiency is slightly better than GA, while the improvement in optimization quality is much more significant.

In summary, Mask A was optimized using five different algorithms—GA, GA-TS, GA-DE, GA-PSO, and GA-HO—and each algorithm was run ten times. Figure 16 shows the comparison of their MR and ERR results across the ten runs. Here, ERR represents the error between the developed pattern and the target design, which reflects how much the final imaging result deviates from the intended mask pattern.

As shown in Figure 16 and Table 2, from the MR and ERR comparison, it is clear that GA-HO performs much better than the other four algorithms. It not only achieves higher matching rates across all ten runs, but also produces noticeably lower ERR values, meaning the final imaging result is closer to the target design. In contrast, the other algorithms either converge more slowly, show limited improvement in the later stages, or have larger fluctuations between runs. GA-HO, on the other hand, finds a better balance between global search and local refinement, giving more stable optimization results and higher image quality. These outcomes further show that combining GA’s global search ability with HO’s strong local adjustment makes GA-HO more effective for complex mask optimization tasks.

Overall, GA-HO clearly performs the best. Its results match the target pattern more closely on the whole, and it also handles the small local details much better. Many of the issues caused by optical proximity effects (OPE)—such as edge shifts, linewidth changes, and rounded corners—are visibly reduced in the GA-HO results.

In contrast, the other algorithms can only fix these OPE problems to a limited degree. Their improvements are much weaker, especially in areas with complex edges or very small features, where noticeable distortion still remains. GA-HO works better because it brings together GA’s global search ability and HO’s strong local adjustment. It can quickly move toward better solutions and then fine-tune the mask more carefully in the later stage, making the final imaging result more stable and reliable.

Overall, the results in the figure show that GA-HO has a clear advantage when dealing with OPE. The improvements are more consistent, the details are handled more accurately, and the final pattern is closer to the target design. This suggests that GA-HO is more adaptable and has greater potential in challenging mask optimization tasks.

Next, the five algorithms were also applied to Mask B, and each algorithm was run ten times. The MR and ERR curves for these runs are shown in Figure 17 and Table 3.

From these results, we can still reach a fairly consistent conclusion. Using GA as the basic reference, GA-TS does run faster than GA, but its main idea is simply adding a tabu list to avoid repeated or clearly useless search directions. In other words, it mainly helps skip unnecessary paths rather than making the algorithm itself “smarter,” so the final optimization quality is almost the same as GA—just with fewer wasted steps.

GA-DE shows a similar situation. Although it is called a “hybrid algorithm,” it does not form a truly complementary structure. GA and DE are already quite similar in nature, as both focus on global search. Putting them together is more like stacking two similar methods rather than letting them enhance each other. Because DE requires extra computation, the overall efficiency becomes lower, and although it can sometimes jump out of local regions and improve the result a bit, the gain is not very large.

GA-PSO works more like a typical two-stage method: the GA is used first to explore the global structure, and PSO takes over later to do some local refinement. This makes the results slightly better than using the GA alone, and the overall behavior is more stable than GA-TS and GA-DE.

Among all the algorithms, the GA-HO algorithm stands out the most. The results show that it performs better not only in overall matching rate but also in fine local details and in correcting distortions caused by optical proximity effects (OPE). HO already has strong local refinement ability on its own, and its leader mechanism keeps guiding the population toward better directions, preventing it from drifting or becoming stuck. This allows the GA-HO algorithm to find a good search direction early on and continue making solid improvements later in the process. In particular, the GA-HO algorithm does a much better job on edge details, far surpassing the other four algorithms.

The five algorithms were also applied to Masks C and D, and each algorithm was run ten times. Figure 18 and Figure 19, along with Table 4 and Table 5, show the MR and ERR curves for these runs.

From Figure 18 and Table 4, it can be seen that the GA-HO hybrid algorithm shows a clear improvement over the other four algorithms, especially in terms of optimization quality.

From the comparison above, it is clear that the GA-HO hybrid algorithm shows a noticeable advantage in both performance and efficiency compared with the other methods. Next, as shown in Figure 20, Figure 21, Figure 22 and Figure 23, this study selects several representative contour results from the ten runs and places them together for a direct comparison.

### 4.3. Analysis of Simulation Results

As shown in the figures from the previous section, the five optimization algorithms behave quite differently when applied to four representative mask structures. Among them, the proposed GA-HO hybrid algorithm stands out with noticeably better overall performance compared to the standard GA and the other four methods. It performs better in terms of optimization efficiency, solution quality, and algorithm stability.

In terms of accuracy, GA-HO maintains high image fidelity even when handling complex masks with dense line structures. Its matching rate (MR) is consistently higher than that of the other algorithms. This shows that GA-HO not only has strong global search capability to explore the solution space effectively, but also performs well in local refinement, leading to more accurate and stable results.

Looking at convergence behavior, GA-TS improves efficiency to some extent by using a tabu list to avoid redundant searches and unnecessary iterations. In contrast, GA and GA-DE lack local refinement mechanisms. Although their convergence appears smooth, they are more likely to become stuck in local optima, resulting in lower-quality results. GA-PSO introduces local search and performs slightly better than GA and GA-DE, but still falls short of GA-HO in terms of convergence speed and overall result quality. In particular, for more complex tasks like OPE correction, HO provides significantly more accurate local refinements than PSO, especially in terms of capturing details and correcting contours. From the data in Table 6, GA-HO clearly outperforms the other methods in matching rate, with improvements of 10 to 30 percentage points across the four masks. Although its runtime is slightly longer than GA-TS due to the added hybrid mechanism, GA-HO is still more efficient overall than GA, GA-PSO, and GA-DE.The key to GA-HO’s improved performance lies in its “local guidance mechanism,” which balances global exploration and local exploitation. It keeps the population diversity and wide search range of GA, while introducing HO’s ability to fine-tune solutions locally, helping the population converge gradually toward better regions and improving both accuracy and efficiency.

In summary, the GA-HO hybrid algorithm delivers excellent results in complex mask optimization tasks. Its combined global-local strategy improves both stability and adaptability, and provides a solid foundation for future high-precision lithography applications. These findings further confirm that GA-HO has strong potential and broad applicability in advanced DMD-based mask design and optimization.

## 5. Conclusions

This study proposes a Genetic–Hippo hybrid optimization algorithm (GA-HO) for DMD-based maskless lithography and shows that it performs significantly better than traditional methods such as GA, GA-TS, GA-DE, and GA-PSO. Experimental results on four representative masks (A-D) demonstrate that GA-HO achieves the highest overall matching rates while keeping runtime at a competitive level.

Specifically, for Mask A, GA-HO reached an average matching rate of 82.39% with a runtime of around 50.1 s, clearly outperforming GA-PSO (68.13%, 53.2 s) and the other algorithms (62–66%). On Mask B, GA-HO also achieved the best result with 81.45% in 51.85 s, while all other methods remained below 65%. For Mask C, GA-HO achieved 81.90% (48.6 s), compared to 75.01% (49.8 s) for GA-PSO and 65–69% for the rest. On Mask D, GA-HO achieved 75.42% (50.86 s), outperforming GA-PSO (65.18%) and the others (58–61%).

The key to this performance improvement lies in the use of a local refinement strategy based on the Hippo optimizer. While GA handles global search to explore the solution space, HO focuses on refining the most promising areas. This combination helps GA-HO converge more smoothly and efficiently while reducing the risk of becoming stuck in local optima. In practice, GA-HO shows more stable convergence and better repeatability. It preserves elite solutions and introduces perturbations when evolution stalls, which helps avoid premature convergence. Overall, GA-HO combines GA’s global search ability with HO’s directional gene updates to achieve fast, reliable, and diverse optimization.

In addition, the masks optimized by GA-HO show better imaging quality and structural accuracy. Comparative results indicate that GA-HO produces more balanced grayscale masks, resulting in sharper edges, clearer contours, and better retention of fine features. It effectively reduces distortions caused by optical proximity effects, such as edge shifts, linewidth errors, and corner rounding. Contour reconstructions further confirm that GA-HO patterns match the target shapes more closely and capture subtle details more accurately. These improvements directly contribute to better results in micro- and nanofabrication.

In summary, GA-HO offers a powerful and scalable optimization solution for high-precision lithography. Its global-local hybrid strategy improves both computation and result quality, and it adapts well to different types of complex masks. Thanks to its modular design, GA-HO can also be extended or integrated into other swarm intelligence frameworks, making it a strong candidate for practical engineering use. This method provides a solid foundation for digital, automated, and high-resolution lithography and shows great promise for advancing maskless lithography technologies.

## Figures and Tables

**Figure 1 micromachines-16-01333-f001:**
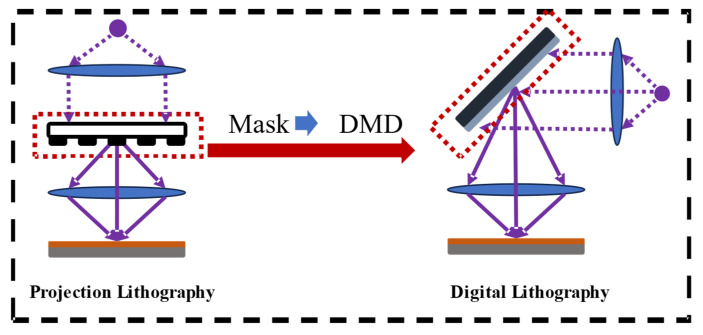
Overview of projection and digital lithography technologies.

**Figure 2 micromachines-16-01333-f002:**
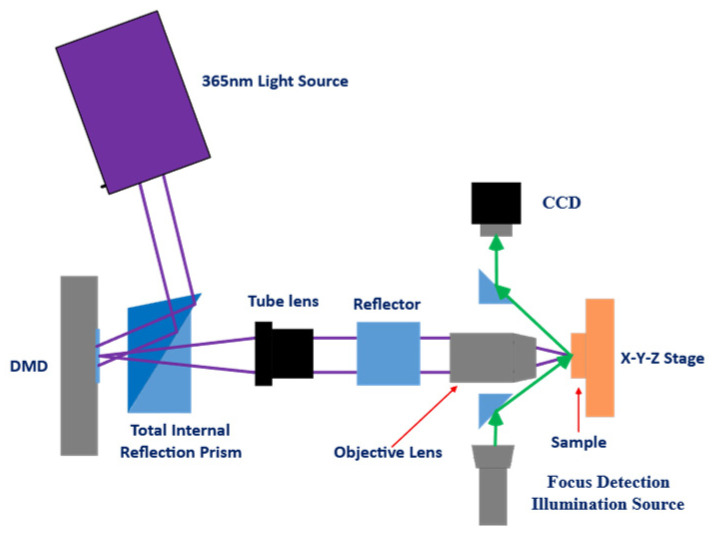
A sketch of DMD-based maskless lithography.

**Figure 3 micromachines-16-01333-f003:**
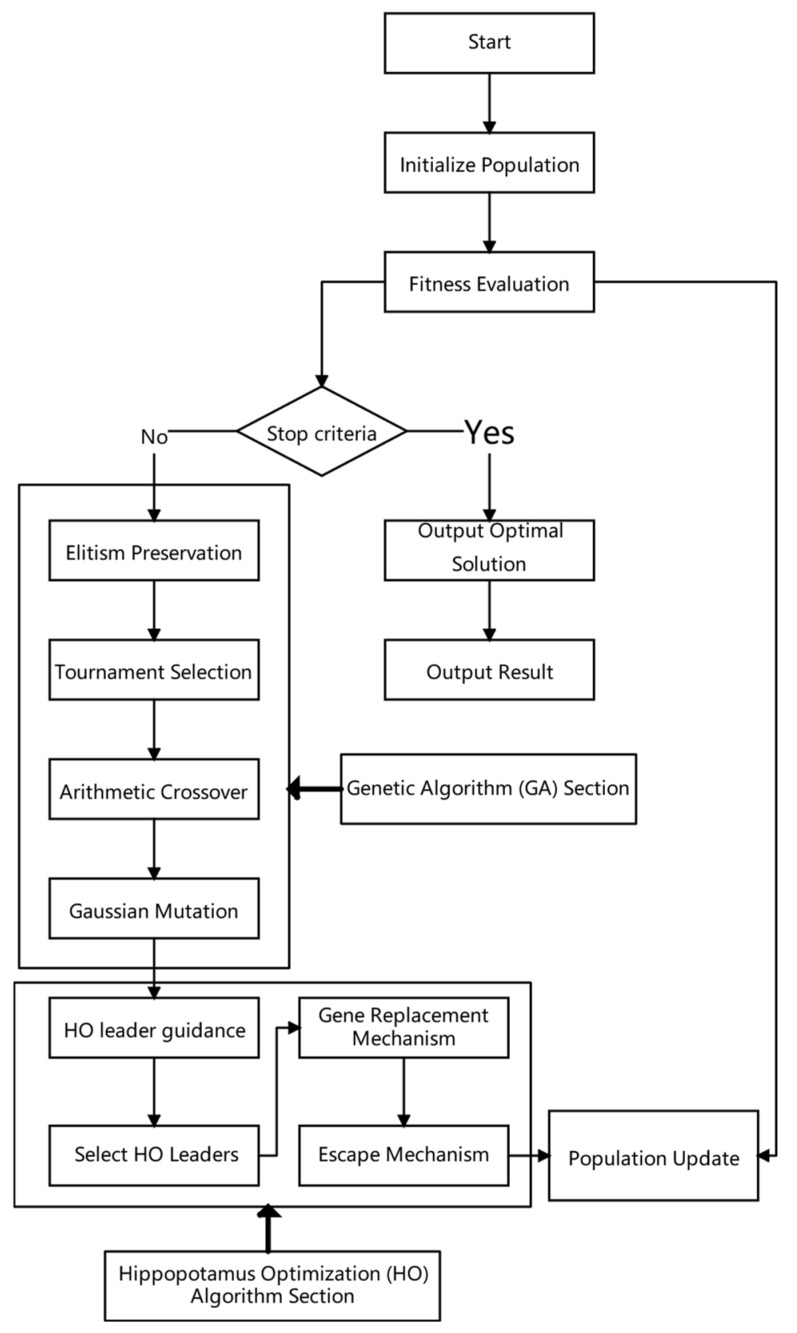
Flowchart of the Genetic–Hippo (GA-HO) Hybrid Optimization Algorithm.

**Figure 4 micromachines-16-01333-f004:**
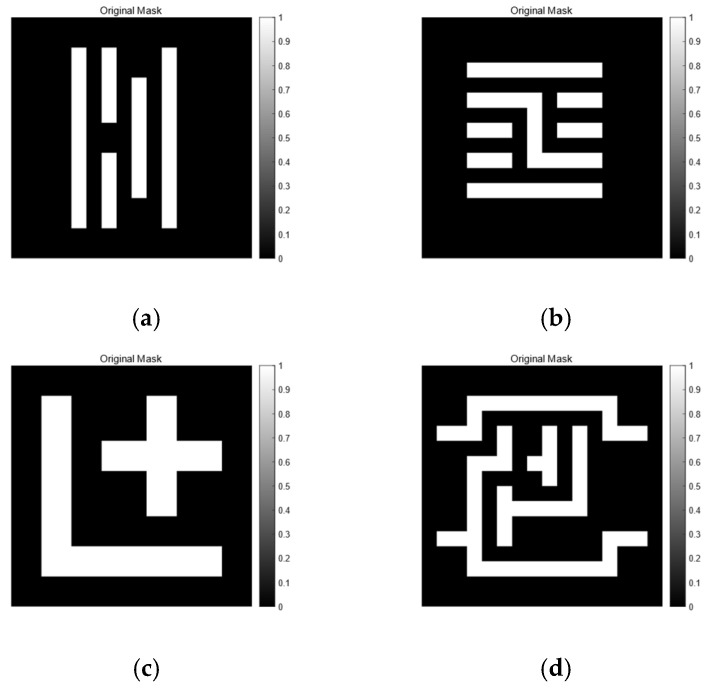
(**a**) shows mask A, with an initial matching rate of 52.7%. (**b**) shows mask B, whose initial matching rate is 50.64%. (**c**) presents mask C, with an initial matching rate of 60.25%. (**d**) shows mask D, which has an initial matching rate of 45.98%.

**Figure 5 micromachines-16-01333-f005:**
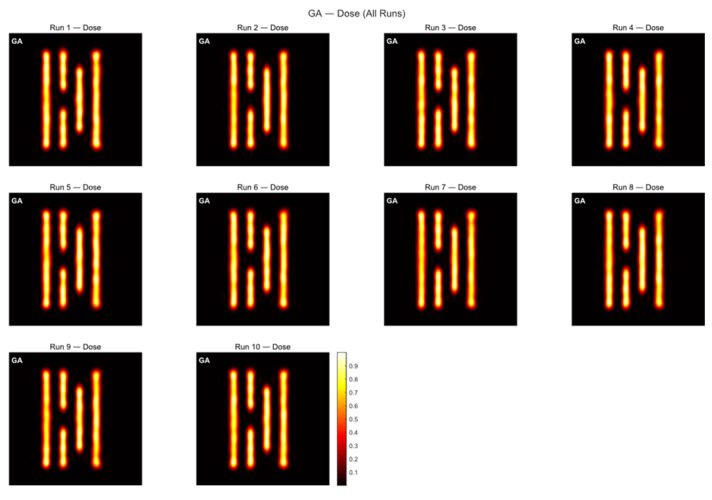
Table exposure patterns obtained by running the GA on Mask A for ten rounds of optimization.

**Figure 6 micromachines-16-01333-f006:**
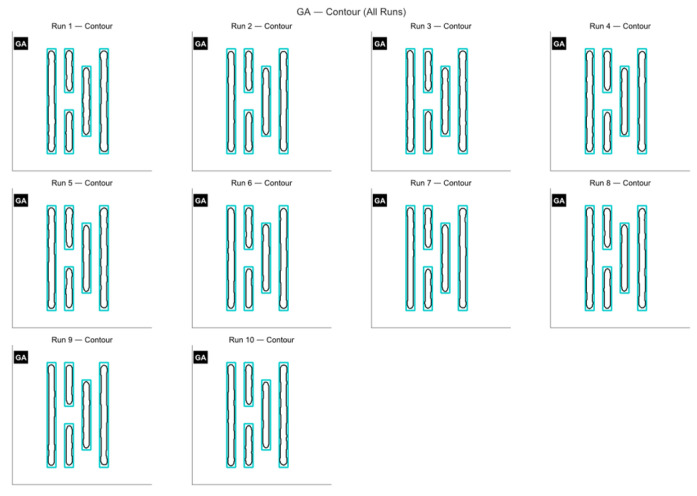
Table contour comparison results obtained by applying the GA to Mask A over ten optimization runs are shown below.

**Figure 7 micromachines-16-01333-f007:**
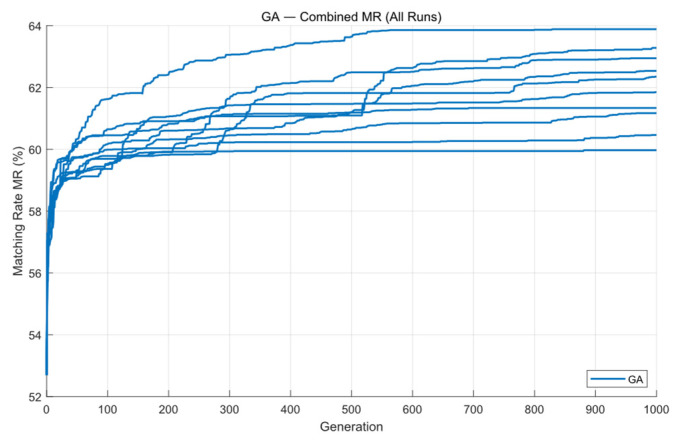
The optimization results for Mask A over ten GA runs.

**Figure 8 micromachines-16-01333-f008:**
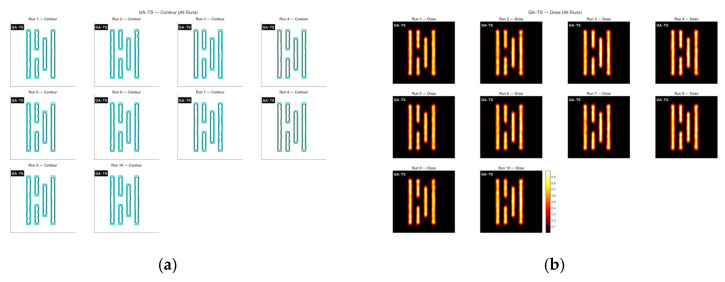
(**a**) Contour comparison showing the original mask in black and the optimized imaging edge in blue; (**b**) exposure patterns from ten GA-TS optimization runs on Mask A. (The black contour represents the original mask, while the blue contour shows the edge of the optimized mask’s imaging result).

**Figure 9 micromachines-16-01333-f009:**
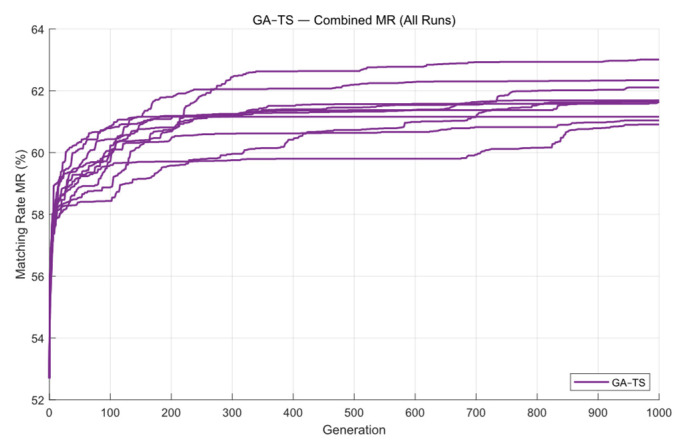
The global MR curves obtained from ten optimization runs of the GA-TS algorithm on Mask A.

**Figure 12 micromachines-16-01333-f012:**
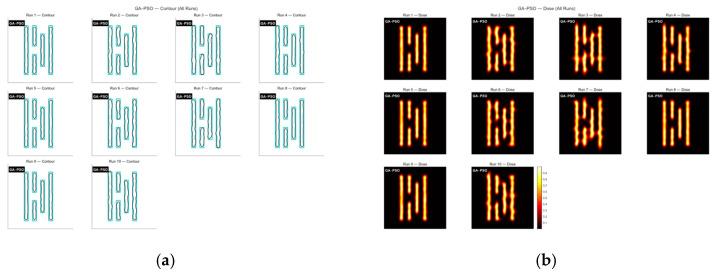
The contour comparison (**a**) and exposure patterns (**b**) obtained from ten optimization runs of the GA-PSO algorithm on Mask A.

**Figure 13 micromachines-16-01333-f013:**
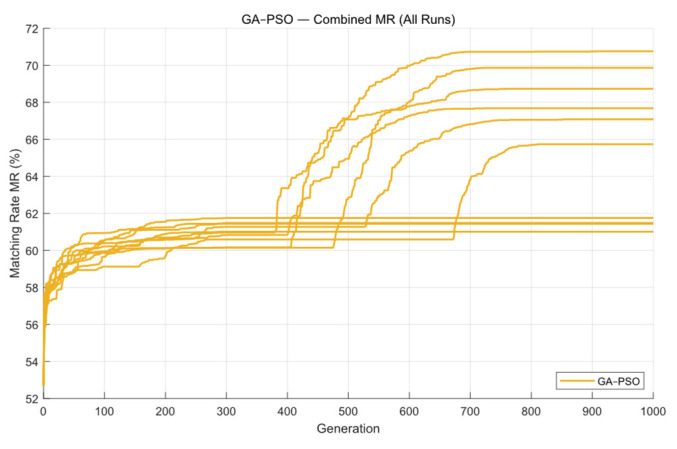
The global MR curves obtained from ten optimization runs of the GA-PSO algorithm on Mask A.

**Figure 14 micromachines-16-01333-f014:**
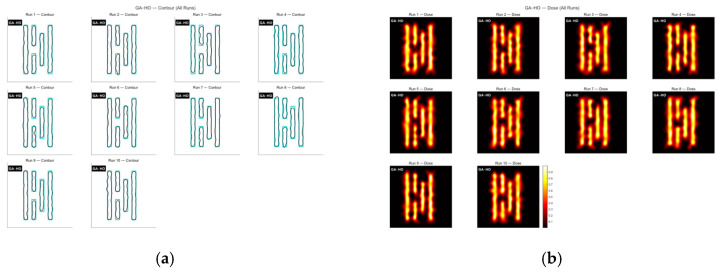
The contour comparison (**a**) and exposure patterns (**b**) obtained from ten optimization runs of the GA-HO algorithm on Mask A.

**Figure 15 micromachines-16-01333-f015:**
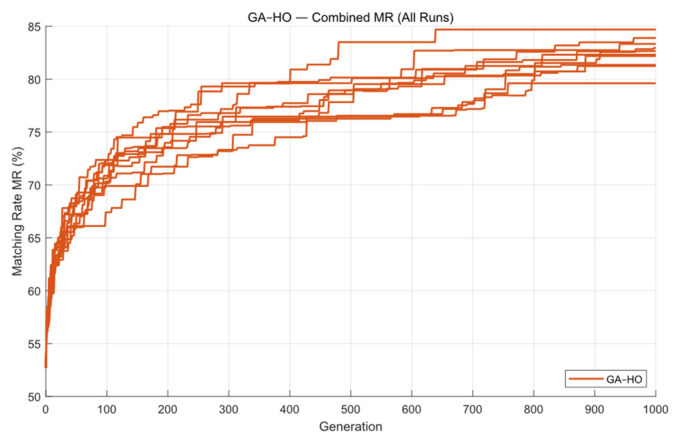
The global MR curves obtained from ten optimization runs of the GA-HO algorithm on Mask A.

**Figure 16 micromachines-16-01333-f016:**
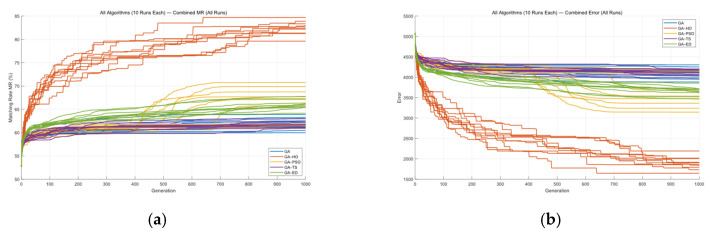
(**a**) MR comparison and (**b**) ERR comparison for Mask A using five algorithms over ten runs.

**Figure 17 micromachines-16-01333-f017:**
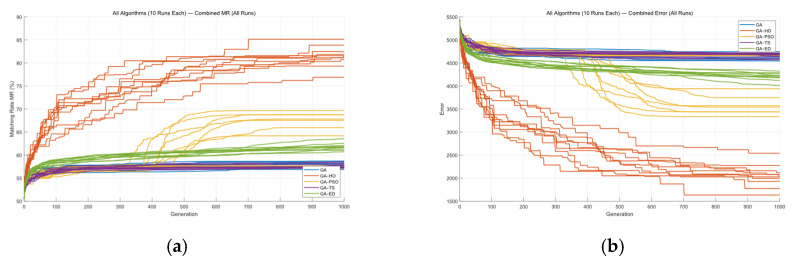
(**a**) MR comparison and (**b**) ERR comparison for Mask B using five algorithms over ten runs.

**Figure 18 micromachines-16-01333-f018:**
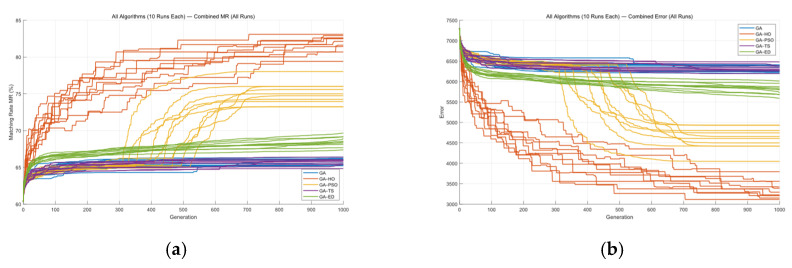
(**a**) MR comparison and (**b**) ERR comparison for Mask C using five algorithms over ten runs.

**Figure 19 micromachines-16-01333-f019:**
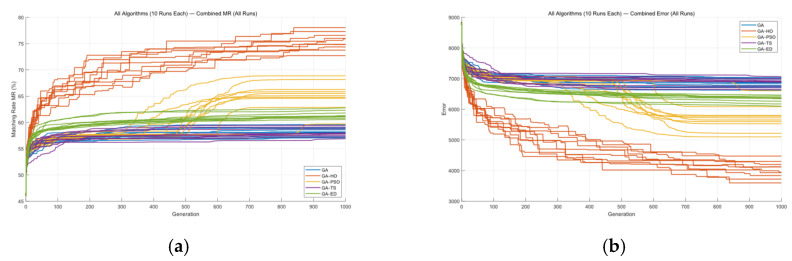
(**a**) MR comparison and (**b**) ERR comparison for Mask D using five algorithms over ten runs.

**Figure 20 micromachines-16-01333-f020:**
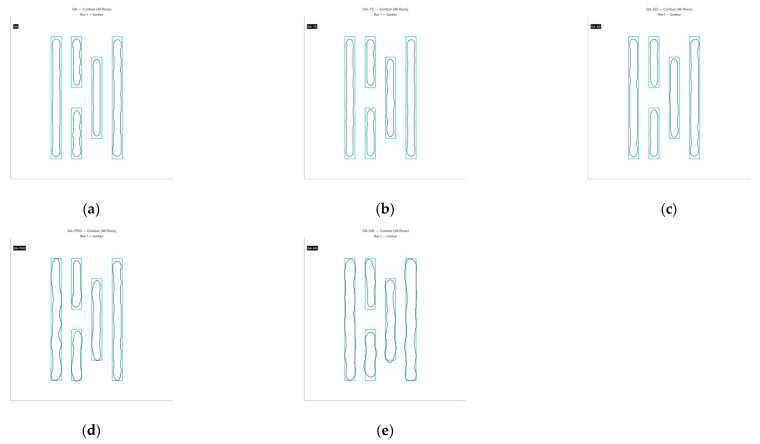
The contour comparison of the original mask and the optimized results for Mask A using the five algorithms is shown below. (**a**) GA, with a matching rate (MR) of 62.65% and an ERR of 4016; (**b**) GA-TS algorithm, with an MR of 62.02% and an ERR of 4084; (**c**) GA-DE algorithm, with an MR of 66.07% and an ERR of 3648; (**d**) GA-PSO algorithm, with an MR of 70.14% and an ERR of 3211; (**e**) GA-HO algorithm, with an MR of 80.78% and an ERR of 2067.

**Figure 21 micromachines-16-01333-f021:**
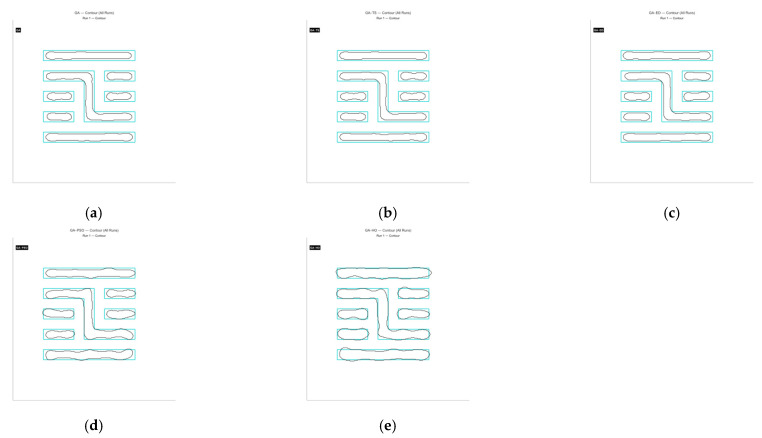
The contour comparison of the original mask and the optimized results for Mask B using the five algorithms is shown below. (**a**) GA, with a matching rate (MR) of 57.14% and an ERR of 4718; (**b**) GA-TS algorithm, with an MR of 57.32% and an ERR of 4719; (**c**) GA-DE algorithm, with an MR of 60.66% and an ERR of 4330; (**d**) GA-PSO algorithm, with an MR of 66.60% and an ERR of 3677; (**e**) GA-HO algorithm, with an MR of 82.29% and an ERR of 1950.

**Figure 22 micromachines-16-01333-f022:**
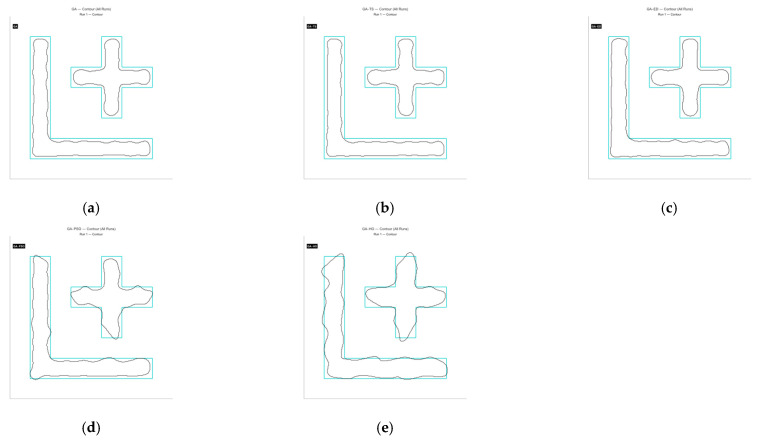
The contour comparison of the original mask and the optimized results for Mask C using the five algorithms is shown below. (**a**) GA, with a matching rate (MR) of 65.65% and an ERR of 6332; (**b**) GA-TS algorithm, with an MR of 65.09% and an ERR of 6434; (**c**) GA-DE algorithm, with an MR of 69.80% and an ERR of 5566; (**d**) GA-PSO algorithm, with an MR of 73.71% and an ERR of 4845; (**e**) GA-HO algorithm, with an MR of 81.47% and an ERR of 3415.

**Figure 23 micromachines-16-01333-f023:**
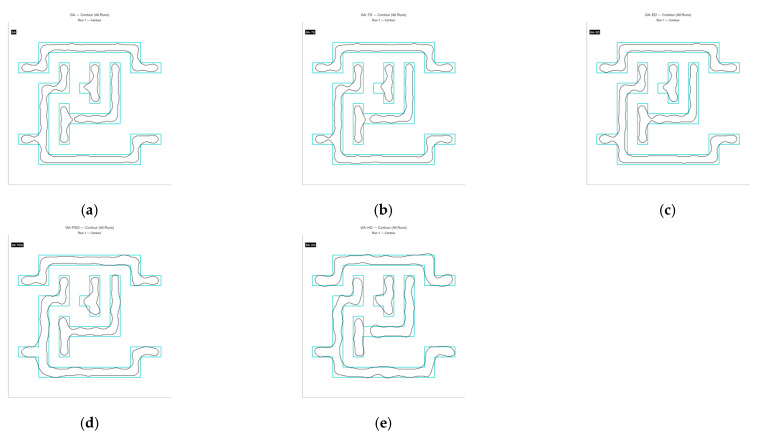
The contour comparison of the original mask and the optimized results for Mask D using the five algorithms is shown below. (**a**) GA, with a matching rate (MR) of 58.45% and an ERR of 6807; (**b**) GA-TS algorithm, with an MR of 58.86% and an ERR of 6223; (**c**) GA-DE algorithm, with an MR of 62.02% and an ERR of 6741; (**d**) GA-PSO algorithm, with an MR of 65.93% and an ERR of 5582; (**e**) GA-HO algorithm, with an MR of 73.89% and an ERR of 4278.

**Table 1 micromachines-16-01333-t001:** Simulation parameter specifications.

Parameter	Value
Wavelength (λ)	365 nm
Numerical Aperture (NA)	0.3
Partial Coherence Factor	0.9
Photoresist Exposure Threshold (T_p_)	0.5
Mask Image Size	16 × 16
Population Size	30
Elite Retention Ratio	0.1
HO Leader Proportion	0.5
Maximum Iterations	1000
Loop	10

**Table 2 micromachines-16-01333-t002:** The average MR, ERR, and runtime for A over ten optimization runs using the five algorithms.

Algorithm	MR	ERR	Time
GA	61.97%	4088	55.31 s
GA-TS	61.72%	4102	48.32 s
GA-DE	66.02%	3657	61.98 s
GA-PSO	68.13%	3302	53.21 s
GA-HO	82.39%	1889	50.12 s

**Table 3 micromachines-16-01333-t003:** The average MR, ERR, and runtime for B over ten optimization runs using the five algorithms.

Algorithm	MR	ERR	Time
GA	57.79%	4646	61.79 s
GA-TS	57.64%	4712	45.62 s
GA-DE	61.91%	4302	63.98 s
GA-PSO	64.51%	3901	52.31 s
GA-HO	81.45%	2015	51.85 s

**Table 4 micromachines-16-01333-t004:** The average MR, ERR, and runtime for C over ten optimization runs using the five algorithms.

Algorithm	MR	ERR	Time
GA	69.77%	6308	62.15 s
GA-TS	65.72%	6702	39.32 s
GA-DE	68.54%	6232	66.98 s
GA-PSO	75.01%	4502	49.77 s
GA-HO	81.90%	3396	48.60 s

**Table 5 micromachines-16-01333-t005:** The average MR, ERR, and runtime for D over ten optimization runs using the five algorithms.

Algorithm	MR	ERR	Time
GA	58.21%	6846	59.38 s
GA-TS	58.04%	5707	45.77 s
GA-DE	61.39%	6336	64.39 s
GA-PSO	65.18%	5702	51.20 s
GA-HO	75.42%	4026	50.86 s

**Table 6 micromachines-16-01333-t006:** For mask types A, B, C, and D, the MR and computation time obtained after optimization using the five algorithms were evaluated.

Algorithm	Comparison Metric	Mask A	Mask B	Mask C	Mask D
/	Original-mask MR	52.70%	50.64%	60.25%	45.98%
GA	MR	61.97%	57.79%	69.77%	58.21%
	Time	55.31 s	61.79 s	62.15 s	59.38 s
GA-TS	MR	61.72%	57.64%	65.72%	58.04%
	Time	48.32 s	45.62 s	39.32 s	45.77 s
GA-DE	MR	66.02%	61.91%	68.54%	61.39%
	Time	61.98 s	63.98 s	66.98 s	64.39 s
GA-PSO	MR	68.13%	64.51%	75.01%	65.18%
	Time	53.21 s	52.31 s	49.77 s	51.20 s
GA-HO	MR	82.39%	81.45%	81.90%	75.42%
	Time	50.12 s	51.85 s	48.60 s	50.86 s

## Data Availability

The original contributions presented in this study are included in the article. Further inquiries can be directed to the corresponding author.

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
