# Peer review of "Efficient Mask Optimization for DMD-Based Maskless Lithography Using a Genetic–Hippo Hybrid Algorithm"

_micromachines, 2025, doi:10.3390/mi16121333_

Round 1

Reviewer 1 Report

Comments and Suggestions for Authors

This manuscript investigates mask optimization as a critical technique for enhancing imaging performance in digital micromirror device (DMD)-based maskless lithography, targeting the challenges of slow convergence, limited adaptability, and reduced imaging accuracy when handling complex multi-feature mask patterns. To overcome these limitations, the authors propose a hybrid Genetic-Hippo Optimization (GA-HO) algorithm that integrates the global exploration capability of Genetic Algorithms (GA) with the local exploitation efficiency of the Hippo Optimization Algorithm (HO). This framework achieves adaptive grayscale mask optimization through a global-local cyclic search mechanism, significantly improving optimization efficiency, convergence stability, and pattern fidelity. The effectiveness of the proposed method is validated via simulation experiments. The research presents solid theoretical foundations and strong potential for practical application. However, the following issues require revision before recommendation for publication:

â‘ Although the authors provided detailed descriptions of the genetic algorithm and hippopotamus optimization algorithm in Sections 3.1 and 3.2 respectively, the specific execution processes of both algorithms have not been shown intuitively. It is recommended that the authors insert an algorithm flowchart in each corresponding section to help readers better understanding the proposed algorithms.

â‘¡The authors previously noted that the proposed GA-HO algorithm exhibits superior convergence speed compared to conventional methods. However, this advantage was not visually shown through charts or data in the simulation experiments. It is recommended to add metrics related to “convergence speed” to Table 2.

â‘¢Although the captions for Figures 6 and 9 mention that they present a comparison between the GA and GA-HO algorithms, the figures themselves do not clearly label which portion represents the GA results and which portion represents the GA-HO results. This lack of labeling may cause confusion for readers.

Reviewer 2 Report

Comments and Suggestions for Authors

The manuscript proposes a hybrid optimization algorithm that combines the Genetic Algorithm (GA) with the Hippo Optimization (HO) algorithm to optimize mask patterns for digital micromirror device (DMD)-based maskless lithography. The authors claim that their GA-HO hybrid approach achieves superior optimization performance compared to the standard GA, with approximately 30% improvement in matching accuracy for complex multi-line mask patterns. While the problem is relevant and the initial results show promise, the manuscript suffers from significant weaknesses that prevent it from meeting the standards for publication. These include insufficient novelty, severely limited experimental validation, oversimplified physical modeling, lack of real experimental verification, and inadequate comparative analysis. I recommend major revisions to address these fundamental concerns.

  1. The core contribution of this paper is the hybridization of two existing metaheuristic algorithms (GA and HO). However, hybrid metaheuristic algorithms are well-established in the optimization literature, and the paper does not provide sufficient theoretical justification for why this particular combination is uniquely suited for DMD mask optimization. Please add the theoretical rationale for combining GA and HO, specifically, supported by mathematical analysis or empirical evidence from preliminary studies. Also, compare the GA-HO approach with other hybrid metaheuristic algorithms (at minimum GA-PSO and GA-DE) to demonstrate its unique advantages.

  1. The most critical weakness of this manuscript is the extremely limited scope of experimental validation. The entire evaluation is based on only two test masks (Mask A and Mask B), both of which are relatively simple patterns of 16×16 pixels. Two test cases are insufficient to demonstrate the generalizability, robustness, and scalability of the proposed method. The patterns tested are simple and may not represent the complexity and diversity of real-world lithography applications. Please expand the test set to include at least 10-20 diverse mask patterns with varying feature sizes, densities, orientations, and complexities.
  2. The analysis of the simulation results is superficial and lacks the depth and rigor expected for a research paper. The convergence curves (Figures 5, 7, 8, 10) appear to be from single runs. No statistical analysis over multiple runs is provided. The paper claims that GA-HO is faster, but no timing data is provided to support this claim.
  3. The manuscript does not provide any comparison with other established algorithms or models. Please include a comparative table summarizing the performance of the proposed GA-HO algorithm against other reported approaches in the literature. This will help substantiate the claimed advantages and position the proposed method within the broader research context.

Round 2

Reviewer 1 Report

Comments and Suggestions for Authors

The revised version answered my question  well and I agree to publish it.

Reviewer 2 Report

Comments and Suggestions for Authors

no further comments.